# Communication patterns in decision-making consultations between patients with advanced cancer and medical oncologists: A qualitative observational study

Daisy J.M. Ermers[1,2]*, Yvonne Engels[1], Henk J. Schers[2], Kris C.P. Vissers[1], Evelien J.M. Kuip[1,3], Marieke Perry[2]

**1** Department of Anesthesiology, Pain and Palliative Medicine, Radboud Institute for Health Sciences, Radboud university medical center, Nijmegen, The Netherlands, **2** Department of Primary and Community Care, Radboud university medical center, Nijmegen, The Netherlands, **3** Department of Medical Oncology, Radboud university medical center, Nijmegen, The Netherlands

* Daisy.Ermers@radboudumc.nl

## Abstract

### Background

Shared decision-making remains underutilized in patients with advanced cancer, despite its proven importance and ongoing efforts to improve its implementation. The influence of communication patterns during consultations on the limited application of shared decision-making in daily clinical practice is not yet well understood. This study explores communication patterns in medical decision-making consultations between patients with advanced cancer and medical oncologists.

### Methods

We conducted a qualitative observational study of single consultations between patients with advanced cancer and their medical oncologists in a Dutch tertiary referral center. We used reflexive thematic analysis to generate key themes and categories that characterize communication patterns during these decision-making consultations.

### Results

From January to March 2019, our analysis of 16 audio-recorded consultations generated four themes. 1. The medical oncologist is balancing between hope and realism. 2. There is little room for bad news. 3. The medical oncologist's medical perspective is leading in medical decision-making. 4. The patient and medical oncologist have a shared focus on anticancer treatment.

**Data availability statement:** The anonymized data underlying the findings are freely available within the manuscript. However, the raw qualitative data (full transcripts) cannot be shared publicly, as doing so would compromise participant (patients' as well as medical oncologists') confidentiality and privacy. Consent for public sharing of raw data was therefore not obtained. Where possible, raw qualitative data will be further anonymized through data aggregation to enable reuse. The anonymized data used for analysis are available from the corresponding author and the research unit in CC (Onderzoek.anes@radboudumc.nl) upon reasonable request.

**Funding:** This study was financially supported by the Radboud University Medical Centre (https://www.radboudumc.nl) in the form of a grant (RvBI7.52147) received by DE. No additional external funding was received for this study. The funder had no role in study design, data collection and analysis, decision to publish, or preparation of the manuscript.

**Competing interests:** The authors have declared that no competing interests exist.

## Conclusions

In consultations between patients with advanced cancer and their medical oncologist, a self-reinforcing cycle of balancing hope and realism alongside a focus on life-prolonging anticancer treatment appears to impede the practice of shared decision-making.

## Introduction

Patients with advanced cancer often face complex treatment decisions, weighing the potential benefits of palliative anticancer treatment against the possible harms [1–4]. In guiding these medical decisions, it is essential to consider the likelihood of treatment response, risk of toxicity, and the patient's personal values and preferences [5]. Shared decision-making has been developed as a strategy to help patients and clinicians align treatment decisions with what matters most to the patient [6].

Several models of shared decision-making have been proposed to support this process. A widely used framework is that of Elwyn et al [6], which distinguishes three key steps: *choice talk* (making clear that a decision exists), *option talk* (presenting available options and their pros and cons), and *decision talk* (supporting patients in exploring preferences and reaching a decision). Stiggelbout et al [7]. have, amongst others, further refined this model by emphasizing deliberation about patient preferences and the patient's preferred role in decision-making. Despite the availability of such theoretical frameworks, the application of shared decision-making in daily oncology practice remains limited. [1,8–10]

Clear evidence supports the importance of shared decision-making [11–13], and it has been integrated into various clinical guidelines [14], policy documents [15], and medical education programs [16,17]. However, numerous barriers to effective implementation have been identified [10,18,19]. While various initiatives have been proposed to enhance its practice, they often fall short of achieving widespread success [20,21].

Recent insights suggest that a strong emphasis on anticancer treatment significantly influences decision-making in patients with advanced cancer [22–24]. A systematic review with meta-synthesis [22] identified an important driver for this: The Overwhelming Situation of 'No Choice,' in which patients, their close ones, and healthcare professionals often feel compelled to pursue aggressive anticancer treatment options, driven by a mutual imperative to combat cancer. An observational study [23] found that while oncologists provide thorough attention to medical aspects, there is an opportunity to focus more on patients' daily functioning and quality of life. Furthermore, the option of forgoing treatment was frequently described as 'doing nothing', which may benefit from reframing.

Nevertheless, research on how this focus on treatment emerges during consultations between patients and medical oncologists is limited, and communication patterns contributing to this focus are poorly understood. Therefore, we aimed to identify such communication patterns in decision-making conversations between patients with advanced cancer and medical oncologists to enable more person-centered palliative care through improved shared decision-making.



## Methods

### Study design

In this qualitative observational study, we applied reflexive thematic analysis to audio-recorded outpatient consultations of the Medical Oncology department. Reflexive thematic analysis was chosen because it allows for an in-depth exploration of communication processes and the identification of recurring patterns, while acknowledging the active role of the researchers in interpreting meaning. This approach was particularly suited to our aim of understanding how shared decision-making unfolds in consultations with patients with advanced cancer. We used the Consolidated Criteria for Reporting Qualitative Research (COREQ) to guide reporting of this study [25]. In addition, we considered the Reflexive Thematic Analysis Reporting Guidelines (RTARG) to enhance clarity and completeness [26]. The research ethics committee of Radboud University Medical Center judged the study to be fully compliant with the Dutch Medical Research Involving Human Subjects Act (case number 2018–4992).

### Setting and participants

Data were collected at a university medical center in the Netherlands from January 10th to March 26th, 2019. Audio recordings were made of single consultations between a medical oncologist or medical oncology fellow and a patient with advanced cancer, along with their close ones when present, during which MRI/CT scan results and treatment decisions were discussed.

Medical oncologists, medical oncology fellows, and nurses approached eligible patients. These were patients with advanced cancer whose practitioner would not be surprised if they were to die within 12 months (Surprise Question [27] answered with no). Exclusion criteria included age under 18, inability to speak Dutch fluently, and inability to complete a questionnaire. Convenience sampling was employed for participant recruitment. Written informed consent was obtained from all patients and medical oncologists involved.

### Study team and reflexivity

Team members had diverse research and clinical backgrounds, including primary care, palliative care, and medical oncology (investigator triangulation). The team also included a professor in Meaningful Healthcare with expertise in contextual communication and care. Additionally, all team members had prior experience conducting qualitative research in clinical settings. Further information on the study team and reflexivity is provided in Supplementary 1.

We acknowledge that our perspectives influenced the attention to communication patterns. For example, DE had substantial prior experience observing oncology consultations [28] and conducting a qualitative embedded multiple-case study on shared decision-making [24], which guided the reflexive interpretation of the data. Moreover, the team was largely composed of general practitioners and palliative care physicians, for whom communication is a core element of their clinical practice. To account for these influences, we engaged in reflexive practices throughout the analysis, including notetaking during coding, iterative code review, and multiple team discussions. These strategies helped ensure that the resulting themes were grounded in the data rather than solely shaped by researchers' preconceptions.

### Data collection

**Audio recordings and transcriptions.** JvM, a researcher with prior experience as a spiritual caregiver at the department, facilitated audio-recording of the consultations. JvM maintained a non-participatory role to minimize influence on interactions. All recordings were transcribed verbatim and anonymized to protect patient-physician confidentiality and comply with ethical standards.

**Consultation context measures and participant characteristics.** To explore the heterogeneity of our convenience sample, we collected quantitative data describing the participants and the consultations. Consultations were assessed

in terms of the extent of shared decision-making and participants' immediate perceptions. Each consultation was scored by the first author (DE) using the OPTION-12 instrument (Supplementary 2) [29]. Immediately following the consultation, patients and medical oncologists (fellows) completed a brief questionnaire assessing their satisfaction with the consultation, including the extent of shared decision-making. Basic characteristics of both patients and medical oncologists were also collected.

Patient Questionnaires included:

- SDM-Q-9 [30,31] to assess the patient's involvement in decision-making.

- VAS (Visual Analogue Scales) [32] to rate satisfaction with shared decision-making and general satisfaction with the consultation.

Medical Oncologist Questionnaires included:

- VAS scores for satisfaction with shared decision-making and general satisfaction with the consultation.

- Estimates of patient satisfaction with decision-making and general satisfaction with the consultation.

Participant Characteristics documented were:

- Patients: gender, age, type of cancer, educational level [33], and marital status.

- Medical oncologists: gender and work experience.

### Data analysis

**Content of the consultations.** Transcripts were imported into ATLAS.ti (version 23) for reflexive thematic analysis [34]. DE read all transcripts and inductively coded concepts as closely as possible to the participant's words to minimize subjectivity. Between October 2023 and February 2024, DE and MP iteratively discussed the codes during weekly meetings until consensus was reached. If new codes emerged, all consultations were reviewed in relation to the latest codes. Next, DE and MP grouped similar concepts into initial categories and themes, resulting in a preliminary codebook. During the first research meeting, the authors reviewed the codes to ensure they reflected the data closely, with minimal interpretation or judgment. In the second meeting, the authors similarly ensured that the creation of categories and themes did not introduce excessive interpretation, and consensus on the final codebook was reached with the full team. Investigator triangulation helped to ensure that the themes reflected the full range and depth of the data [35]. Additionally, we evaluated the consistency of constructed themes across consultations (Supplementary 3) and found that each theme was widely supported, thereby validating the themes created.

**Contextualization of the consultations.** We used descriptive statistics for the questionnaires and patients' and medical oncologists' characteristics. These analyses were performed with IBM SPSS software version 29.

## Results

### Participant characteristics

Sixteen patients, eight medical oncologists and two medical oncology fellows participated. Each patient was accompanied by one or more close ones. Patients had different types of cancer, and their gender, age, educational level, and marital status varied (Supplementary 4). Seven of the medical oncologists and medical oncology fellows were women. The medical oncologists had between 2 and 35 years of experience in the field, while the medical oncology fellows had 6–8 years of overall medical experience.

For clarity, from this point forward, the term "medical oncologists" will be used to refer to both medical oncologists and medical oncology fellows.

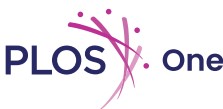

## Consultation context measures

Sixteen consultations were audio-recorded and analyzed. The median duration of the consultations was 20 minutes (range 5–40).

Table 1 displays contextual information of the consultations. In nine consultations, scan results showed disease stability or remission; in one of these, the treatment strategy changed. In five consultations, scan results indicated progressive disease; the treatment strategy changed in two, while in three it was postponed pending a multidisciplinary meeting or further diagnostics. In two consultations, scan results were ambiguous regarding disease stability or progression; in one case the treatment strategy remained unchanged, and in the other, the decision was postponed. Supplementary 5 provides an overview of consultation content.

OPTION-12 scores were generally low, regardless of whether a treatment decision was made. Patient-reported satisfaction was generally high, including satisfaction with the extent of shared decision-making. Just in one consultation (No. 9), the patient reported lower satisfaction. In this case, the medical oncologist conveyed disease progression and advised against starting immunotherapy.

## Communication patterns in decision-making conversations

We generated 118 codes, grouped into 48 axial codes, 15 categories, and 4 themes (Supplementary 6). The themes and categories are described in the following sections, and illustrative quotations per category are provided in Table 2.

**Theme 1: The medical oncologist is balancing between hope and realism.** During several consultations, we observed that bad news—such as disease progression, risk of recurrence, or limited life expectancy—was delivered with a balanced or optimistic tone. For example, one medical oncologist expressed hope before delivering bad news and several highlighted the positive effects of ongoing anticancer treatment or the availability of further treatment options afterward. This also happened when addressing patient concerns. Conversely, if a patient or close one appeared overly optimistic about their prognosis or treatment outcomes, the medical oncologist seemed to manage their expectations accordingly.

**Theme 2: There is little room for bad news.** In several consultations involving bad news, the information was not always explicitly addressed. In some cases, the bad news was softened or followed by a change in subject. This softening involved medical oncologists using more gentle language, such as diminutives like 'tiny gland' or euphemisms. Additionally, some medical oncologists used many words or medical jargon when providing explanations, which could complicate the patient's understanding of the message. In one case (Consultation No. 5), this appeared to lead to the patient drawing their own conclusion about the bad news (Table 2). In some consultations, patients asked questions later, suggesting that the bad news shared at the start had only just been processed. For example, patients asked questions about scan results and disease progression after the medical oncologist had already proceeded to another subject or at the end of the consultation.

Patients and medical oncologists sometimes seemed to provide little space for negative emotions. In some cases, medical oncologists redirected the discussion or rephrased patient statements when concerns were expressed.

**Theme 3: The medical oncologist's medical perspective is leading in medical decision-making.** The medical oncologist predominantly discussed medical topics, including pathophysiology, diagnostic imaging, clinical outcomes, and treatment efficacy. In several consultations, the patient addressed non-medical subjects, such as personal values, preferences, and quality of life concerns. This suggests that the perspectives of the medical oncologist and patient differ. Consultation No. 6 further illustrates this difference, as the medical oncologist responded to the patient's psychoemotional and existential question with factual medical information.

Decision-making appeared to be primarily guided by the medical oncologist, who often suggested treatment options, made decisions, and requested consent without incorporating the patient's perspective. This seemed to be further compounded by a frequent lack of structure in consultations. For instance, in consultation No. 2, the medical oncologist suggested continuing treatment during a physical examination (Table 2).



**Table 1. Contextual information of the consultations.**

| | | Consultation | | | | | | | | | | | | | | | |
|---|---|---|---|---|---|---|---|---|---|---|---|---|---|---|---|---|---|
| | | 1 | 2 | 3 | 4 | 5 | 6 | 7 | 8 | 9 | 10 | 11 | 12 | 13 | 14 | 15 | 16 |
| **Scan result[1]** | | + | + | + | − | − | − | − | + | − | + | + | +/- | + | + | +/- | + |
| **Change in treatment strategy** | | − | − | − | −[2] | −[2] | + | + | − | −[2] | − | + | − | − | − | −[2] | − |
| **OPTION-12 item** | 1 | − | − | − | + | + | +/- | +/- | − | +/- | − | − | − | − | − | +/- | − |
| | 2 | − | − | − | +/- | + | − | − | − | − | − | − | − | − | − | − | − |
| | 3 | − | − | − | − | − | − | − | − | − | − | − | − | − | − | − | − |
| | 4 | − | − | − | +/- | + | − | +/- | − | − | − | − | − | − | − | − | − |
| | 5 | − | − | − | +/- | +/- | +/- | +/- | − | +/- | − | − | − | − | − | +/- | − |
| | 6 | − | − | − | − | − | − | +/- | − | − | − | − | − | − | − | − | − |
| | 7 | − | − | − | − | − | − | − | − | − | − | − | − | − | − | − | − |
| | 8 | − | − | − | + | − | +/- | − | − | +/- | − | − | − | − | − | − | − |
| | 9 | − | − | − | + | − | − | − | − | − | − | − | − | − | − | − | − |
| | 10 | − | − | − | − | − | − | − | − | − | − | − | − | − | − | − | − |
| | 11 | − | − | − | − | − | − | − | − | − | − | − | − | − | − | − | − |
| | 12 | − | − | − | − | − | − | − | − | − | − | − | − | − | − | − | − |
| **SDM-Q-9[3]** | My doctor made it clear that a decision must be made | --- | --- | --- | +++ | +++ | +++ | ++ | +++ | +++ | --- | +++ | --- | --- | + | +++ | + |
| | My doctor wanted to know exactly how I want to be involved in making the decision | --- | --- | --- | +++ | +++ | +++ | ++ | +++ | ? | --- | +++ | --- | --- | --- | +++ | + |
| | My doctor told me that there are different options for treating my condition | +++ | --- | --- | +++ | +++ | +++ | +++ | --- | +++ | --- | +++ | -- | --- | --- | ++ | --- |
| | My doctor precisely explained the (dis)advantages of the treatment options | ++ | --- | --- | +++ | +++ | +++ | +++ | --- | − | --- | +++ | --- | --- | --- | + | + |
| | My doctor helped me understand all the information | +++ | --- | +++ | +++ | +++ | +++ | ? | +++ | − | +++ | +++ | +++ | +++ | --- | +++ | +++ |
| | My doctor asked me which treatment option I prefer | − | --- | --- | +++ | +++ | +++ | +++ | --- | +++ | --- | +++ | --- | --- | --- | +++ | − |
| | My doctor and I weighed the different treatment options thoroughly | + | --- | --- | +++ | +++ | +++ | +++ | --- | -- | --- | +++ | --- | --- | --- | ++ | + |
| | My doctor and I selected a treatment option together | +++ | --- | --- | +++ | +++ | +++ | +++ | --- | --- | --- | +++ | --- | --- | --- | ++ | + |
| | My doctor and I came to an agreement on how to proceed | +++ | --- | +++ | +++ | +++ | +++ | +++ | +++ | + | +++ | +++ | +++ | +++ | ++ | +++ | ++ |
| **VAS on patient satisfaction** | How satisfied are you with your involvement in the decision-making during this consultation? | 85 | 95 | 100 | 95 | 100 | 100 | 80 | 100 | 5 | 100 | 100 | 100 | NA | 80 | 95 | 100 |
| | How satisfied are you with the conversation as a whole? | 95 | 95 | 100 | 95 | 95 | 100 | 95 | 100 | 90 | 100 | 95 | 100 | 85 | 95 | 100 | 100 |
| **VAS on medical oncologist satisfaction** | How satisfied are you with the application of decision-making during this consultation? | 70 | NA | NA | 75 | 20 | 90 | 90 | 80 | 15 | 85 | 95 | 80 | − | 75 | 70 | 75 |
| | How satisfied are you with the conversation as a whole? | 70 | 70 | 55 | 70 | 60 | 90 | 80 | 75 | 65 | 85 | 95 | 80 | − | 75 | 70 | 75 |
| | How satisfied do you think the patient is with their involvement in the decision-making during this consultation? | 75 | NA | NA | 70 | 80 | 90 | 85 | 80 | 5 | 100 | 100 | 80 | − | 65 | 80 | 75 |

*(Continued)*



**Table 1.** (Continued)

| | | Consultation | | | | | | | | | | | | | | |
|---|---|---|---|---|---|---|---|---|---|---|---|---|---|---|---|---|
| | | 1 | 2 | 3 | 4 | 5 | 6 | 7 | 8 | 9 | 10 | 11 | 12 | 13 | 14 | 15 | 16 |
| | *How satisfied do you think the patient is with the conversation as a whole?* | 70 | 75 | 65 | 70 | 65 | 85 | 80 | 80 | 10 | 100 | 100 | 80 | – | 70 | 80 | 75 |

[1]Disease status is stable or in remission: +; disease is progressing: -; ambiguous regarding whether the disease is stable or progressing: +/-

[2]Treatment decision was postponed until after a multidisciplinary meeting or diagnostics

[3]completely disagree: ---; strongly disagree: --; somewhat disagree: -; somewhat agree: +; strongly agree: ++; completely agree: +++

**Table 2.** Themes, categories, and illustrative quotations.

| Themes and categories | Consultation Nos. and illustrative quotations |
|---|---|
| The medical oncologist is balancing between hope and realism | |
| *The medical oncologist presents bad news positively* | Consultation No. 11<br>Medical oncologist: *"And then we hope that it lasts as long as possible. Um … yes, I hope for several months and hopefully even longer. But, yes, eventually it will become active again, and if you are still as active as you are now, well, then we will need to see if you can receive immunotherapy."* |
| | Consultation No. 3:<br>Medical oncologist: *"Yes, but I will be honest with you: once it comes back, it will always come back at some point."*<br>Patient: *"It can always happen."*<br>(…)<br>Medical oncologist: *"And we just have to wait and see what happens, right? If the disease comes back, then we need to determine what the best treatment for you is."*<br>Patient: *"Yes."*<br>Medical oncologist: *"And there are still many treatments available, but you should be aware: it will come back eventually."* |
| | Consultation No. 4<br>Medical oncologist: *"What we see now is that – essentially – everything is still the same, but one lymph node in the chest cavity, in the middle, has indeed grown. So that is – in fact – quite disappointing."* |
| *The medical oncologist responds positively to the patient's negative emotion* | Consultation No. 16<br>Patient: *"I hope I can make it."*<br>Medical oncologist: *"Well, so far, this all looks good. So, it's working."*<br>Patient: *"Yeah, as long as this is good, let's keep going."*<br>Medical oncologist: *"This treatment could be effective for a year and a half. So, if all goes well…"*<br>Patient: *"If it goes well, but in the meantime…"*<br>Medical oncologist: *"So, we're keeping all options…"*<br>Patient: *"You're still trying everything."*<br>Medical oncologist: *"Exactly. We're keeping all options open and there are indeed many new treatments coming. PSMA therapy is one of them, but there's also a treatment with Olaparib, another new drug, that might be just as good or even better than the PSMA therapy. So, there are many new things on the horizon."*<br>Patient: *"Well, yeah."*<br>Medical oncologist: *"So, that's…"*<br>Patient: *"We'll wait and see, and I hope I can make it."* |

*(Continued)*



| Themes and categories | Consultation Nos. and illustrative quotations |
|---|---|
| *The medical oncologist provides the option of anticancer treatment (even if it contradicts own insights)* and *The medical oncologist and patient jointly navigate expectations* | Consultation No. 9<br>Medical oncologist: *"(…) But to be honest, listening to you and looking at you, <u>I wonder if immunotherapy will help you because your condition is such that I'm concerned the cancer might be too advanced for immunotherapy to make a difference…</u>"*<br>Close one: *"Yeah … But what if you start it anyway?"*<br>Medical oncologist: *"One needs to have a certain condition to expect a benefit from immunotherapy. From what I hear, you're mostly just lying on the couch…"*<br>Close one: *"Yeah, the last few weeks."*<br>Medical oncologist: *"Yes. That's a sign of how aggressive the disease is. Cancer consumes energy, both literally and figuratively. The fact that you feel so unwell is, I think, a sign of how aggressive the disease is. We know that if the cancer is in the liver, it makes the chance of immunotherapy <u>working even smaller</u>…"*<br>Close one: *"But we should try all the chances we have."*<br>Medical oncologist: *"I understand that it feels that way, wanting to try everything … But it's a treatment that can also have side effects. I don't want to make you sicker. And seeing how much effort it takes for you to come to the hospital, I understand that you're willing to do it, but I also want you to have the time you have at home, not just in a taxi to the hospital, especially when the chance of success is <u>very low</u>…"*<br>Close one: *"But you have to take that chance, right? You should at least try?"*<br>Medical oncologist: *"Well, I think you need to carefully consider whether … You should only pursue a treatment if you expect it to be meaningful or if there's a realistic chance of success, or if the overall burden is worth it. And I'm not just talking about the financial costs. I'm referring to the effort your wife has to make to come to the hospital, the strain of traveling to [name city]. And we haven't even talked about the possible side effects yet. It's about weighing the costs against the potential benefits."*<br>Close one: *"Yes. Yes."*<br>Medical oncologist: *"<u>I don't expect it to be benefit your wife.</u> Can I guarantee that? No (…)."*<br>(…)<br>Medical oncologist: *"Yes, I understand your desire to do something, but as doctors, we also need to consider whether it's truly worthwhile. I want to make this decision with input from several colleagues and involve your GP as well."*<br>(…)<br>Medical oncologist: *"May I suggest something?"*<br>Patient and close one: *"Yes."*<br>Medical oncologist: *"I'd like to have you weighed and measured shortly. We'll do some blood tests to see if it's even possible … I'll take a more comprehensive blood sample. If my colleagues decide to proceed, or if I see that it's truly feasibly, certain blood values need to be in order. Then I'll discuss with my team how other view the situation. I'll also call your GP, and then I'll call you this week to discuss the results. I'm concerned that my colleagues might also say that immunotherapy might not be feasibly. But I want to discuss it with all my colleagues. If we proceed with this, we all need to be on board. They might say, 'We should give it a change,' and if so, we'll need to start the process and you'll get explanations. …. That means, if we activate your immune system, it could lead to more side effects. It might worsen your kidney function, cause inflammation of the intestines, liver, or lungs. Fortunately, we don't see these issues very often, but if you experience them, you'd become very ill. I worry that you might become even sicker than you are now and I'm really concerned whether you could handle that.*<br>Close one: *"Yeah … What should I say? Nothing ventured, nothing gained (In Dutch: nee heb je, ja kun je krijgen)."* |
| There is little room for bad news | |
| *After discussing bad news, the medical oncologist and patient abruptly change the subject* | Consultation No. 5<br>Medical oncologist: *"Are you afraid that it might come back?"*<br>Patient: *"No, not afraid, but it's certainly not pleasant."*<br>Medical oncologist: *"It's not nice to know, that's true. (Typing on keyboard) May I examine you?"* |
| | Consultation No. 6 (medical oncologist and patient briefly discussed end-of-life wishes)<br>Patient: *"When the time comes, I'll just go, right?"*<br>Medical oncologist: *"When the time comes."*<br>Patient: *"But first, we're going to Texel (A Dutch Island)."* |
| *The medical oncologist does not acknowledge the patient's negative emotions* | Consultation No. 14<br>Patient: *"Yeah. If things go well for a long time, I sometimes think 'Yeah, at some point … (it will come back)'"*<br>Medical oncologist: *"Not necessarily. Not necessarily."* |
| | Consultation No. 9<br>Patient: *"I'm only 71. (Crying) We had a good life together."*<br>Medical oncologist: *"Have a good life together."* |

*(Continued)*

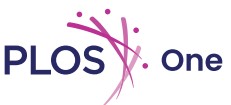

**Table 2.** (Continued)

| Themes and categories | Consultation Nos. and illustrative quotations |
|---|---|
| *The language used by the medical oncologist masks bad news* | Consultation No. 11<br>Medical oncologist: *"Yes, of course, and I don't want to keep rubbing it in, but we did honestly say to each other: "I hope to keep it stable for as long as possible." You then said to me: "I am aiming for cure." And I said "Well, that I can't promise.""*<br>Patient: *"No."*<br>Medical oncologist: *"So, I'm keeping it as stable as possible, but actually, it is more about extending life, isn't it? The treatment is not…"*<br>Patient: *"Fortunately."*<br>Medical oncologist: *"Yes, we probably can't get rid of it entirely."* |
| *The patient doesn't understand the medical oncologist's message due to indirect communication* | Consultation No. 5<br>Medical oncologist: *"We made an ultrasound of those tiny glands. And we did a virus serology because they were so small that we thought, 'They could possibly also be related to an infection.'"*<br>Patient: *"HmmHmm."*<br>Medical oncologist: *"Well, you have indeed had a few viral infections in the past, but nothing very active, so that can't be the explanation. What they see on the ultrasound is that the architecture of the lymph nodes is disrupted."*<br>Patient: *"HmmHmm."*<br>Medical oncologist: *"And that means it's very possible that it could be..."*<br>Patient: *"Cancer."*<br>Medical oncologist: *"malignant cells again."* |
|  | Consultation No. 9<br>00.35.19 Medical oncologist: *"(…) Do you agree that we arrange it this way?"*<br>Patient: (Crying) *"Yes. It is what it is."*<br>Medical oncologist: *"Yes."*<br>Patient: (Crying) *"How long do I have left if it can't be done? I don't think it will be long."*<br>Medical oncologist: *"No, I think so too. But honestly, if I believed that we could significantly extent your time with immunotherapy, I would start tomorrow. But, I don't think we can. I believe it's better not to use the time you have left on treatments that we don't think will be effective. I understand this is very difficult. I want to discuss this with my colleagues to get their opinions. I also plan to call your GP to see if he can help make the most of the time you have left with you and your family. I think it's important to focus on quality, as that is the most crucial aspect."*<br>Patient: (Crying) *"How long does it take for the immunotherapy …?"*<br>Close one: *"To take effect?"*<br>Medical oncologist: *"Sometimes it takes nine to twelve weeks. That's why I'm concerned. Given the rapid deterioration of your condition over the past few weeks. I'm worried that nine to twelve weeks might be too long for the immunotherapy to show results. If it even does, as the chances are not very high. That's my assessment."* |
| The medical oncologist's medical perspective is leading in medical decision-making |  |
| *The medical oncologist's and patient's perspectives differ: medical vs non-medical* | Consultation No. 6<br>Patient: *"Will I make it to my birthday?"*<br>Medical oncologist: *"Yeah, I think so."*<br>Close one: *"It's also about how it happens."*<br>Medical oncologist: *"I'd find that really strange (read: if you didn't make it till June)."*<br>Patient: *"I live from Christmas to June, and in June it's my birthday. I keep saying 'And now it's back to Christmas again'."*<br>Close one: *"No, you don't know yet."*<br>Medical oncologist: *"Look, I don't know how much you know about what was there before, but now it's this."*<br>(Then the medical oncologist, patient and close one review the scan results.) |
|  | Consultation No. 4<br>Patient: *"Yeah. We just need to do what's best."* (During the consultation it remains unclear what's best)<br>Medical oncologist: *"Yeah."*<br>Close one: *"Right, what's important is… Let's say if radiotherapy can be done, it should be … Yeah done as safely as possible, of course."*<br>Patient: *"Definitely."*<br>Close one: *"What risks are involved."*<br>Medical oncologist: *"You always must weigh the risks and benefits. And not in terms of money, but what it costs in terms of side effects."* |

*(Continued)*



**Table 2.** (Continued)

| Themes and categories | Consultation Nos. and illustrative quotations |
|---|---|
| *The medical oncologist and patient do not discuss the patient's context in relation to decision-making* | Consultation No. 1<br>Medical oncologist: *"So, I'll print out a new prescription, and then I'll schedule an appointment in about four months. When are you heading back to Texel (A Dutch Island)?"* |
| | Consultation No. 13<br>Medical oncologist: *"So, what are you up to these days?"*<br>Patient: Laughing) *"What am I up to these days? Physical therapy, three times a week."*<br>Medical oncologist: *"Okay."*<br>Patient: *"Uhm … I'm finishing up my thesis."*<br>Medical oncologist: *"Oh?"*<br>Patient: *"Yeah."*<br>Medical oncologist: *"On what topic?"* |
| *The medical oncologist leads decision-making* | Consultation No. 2<br>Medical oncologist: *"Okay, well, if you don't have any more questions, I'll just examine you and, um …"*<br>Patient: *"Yes, that's fine."*<br>Medical oncologist: *"… we'll continue with this treatment. (Curtain is opened and closed) Was it easy to get here in this weather?"* |
| *The patient is willing to be involved in the decision-making* | Consultation No. 7<br>Patient: *"Yeah. That was my suggestion, to start the radiotherapy."*<br>Medical oncologist: *"Yeah."* |
| | Consultation No. 16<br>Patient: *"Yeah. And I think… if I have to get a new tube put through the urethra every six weeks, I'd rather just have it removed altogether."* |
| The patient and medical oncologist have a shared focus on anticancer treatment | |
| *The medical oncologist and patient reinforce each other's focus on anticancer treatment* | Consultation No. 4<br>Patient: *"Yeah, what needs to be irradiated, must be irradiated."*<br>Medical oncologist: *"Yeah."*<br>Patient: *"No choice, really."*<br>Medical oncologist: *"Yeah (…) Well. You mainly took notes, huh? There aren't any questions left on that page?"* |
| | Consultation No. 5<br>Patient: *"Okay. (…) Yeah, it must be done – there's no other way."*<br>Medical oncologist: *"Well, you always have a choice."*<br>Patient: *"Yeah, but still."*<br>Medical oncologist: *"… but I think it's the wise choice."* |
| | Consultation No. 9<br>Medical oncologist: *"(…) Do you agree that we arrange it this way?"*<br>Patient: (Crying) *"Yes. It is what it is."*<br>Medical oncologist: *"Yes."* |
| *The medical oncologist and patient attest to the positive effect of anticancer treatment on disease progression* | Consultation No. 5<br>Medical oncologist: *"I thought to myself, 'Well, that's one of my little miracles."* |
| | Consultation No. 11<br>Medical oncologist: *(…) And you've shown that you can handle it (chemotherapy). So you can be proud of that."*<br>Patient: *"Guys, you hear that? Thank you doctor. I'm really happy about it."*<br>Medical oncologist: *"So, you did a great job. Really. Better than expected."* |
| | Consultation No. 13<br>Medical oncologist: *"It's a.. It's a little miracle."*<br>Patient: *"That's something you didn't expect, I think?"*<br>Medical oncologist: *"Well, that's what you hope for. That's what we've been working towards."*<br>Patient: *"Yeah."*<br>Medical oncologist: *"But it was really nerve-wracking (…)"*<br>Patient: *"Yeah, just bizarre."*<br>Medical oncologist: *"This is a moment to celebrate."*<br>Patient: *"Yeah, absolutely."* |

*(Continued)*



**Table 2.** (Continued)

| Themes and categories | Consultation Nos. and illustrative quotations |
|---|---|
| *The medical oncologist and patient or their close ones try to convince each other of the value of undergoing/continuing treatment in case of medical oncologist's doubts* | Consultation No. 9<br>Medical oncologist: *"Yeah. But the urologist doesn't prescribe it (immunotherapy). I also think, from what I've seen and how the urologist described your condition, that you've really deteriorated lately, haven't you?"*<br>Close one: *"It's actually been worsening lately… since she found out those cells were there…"*<br>Medical oncologist: *"Yes."*<br>Close one: *"… it's gotten really bad. It's just because she's so upset about it."*<br>Medical oncologist: *"That's right."*<br>Close one: *"But it's not because it's gotten worse beyond that."*<br>Medical oncologist: *"No, but the fact that you're getting so little done and saying, "I'm just so quickly exhausted," is a sign of the illness…"*<br>(…)<br>Medical oncologist: *"Yeah. What do you feel, ma'am? Do you understand what I'm saying?"*<br>Patient: *(Crying)* *"Yes, but I still want to try (immunotherapy)."*<br>Medical oncologist: *"Do you also feel like it's really deteriorating, or do you not feel that way?"*<br>Patient: *(Crying)* *"Yes. Yes. I can do less and less. I get very short of breath … I know …"*<br>Medical oncologist: *"Yeah. May I listen to your lungs? I want to see if I hear a lot of fluid or something else."*<br>(Physical examination started.) |

**Theme 4: The patient and medical oncologist have a shared focus on anticancer treatment.** Most consultations indicated that medical oncologists and patients reinforced each other's focus on anticancer treatment. In some consultations, patients or their close ones expressed feelings of limited choice and a sense of necessity to pursue treatment. Medical oncologists responded in various ways: some did not address these expressions directly, while others either affirmed or contradicted them. Positive outcomes of prior anticancer treatment were frequently emphasized, with medical oncologists using terms such as "miracle" or encouraging patients to be proud of their efforts (see Table 2).

**Theme interrelations: A self-reinforcing cycle unveiled.** Our analysis and the combined quotations in Table 2 suggested an interconnection between the four themes, forming a self-reinforcing cycle (see Fig 1). Efforts by medical oncologists to balance hope and realism influenced how bad news was delivered, often diluting the message. This dilution could make it more challenging for patients to engage as equal partners in decision-making. Consequently, the medical oncologist's perspective tended to dominate the conversation, likely limiting consideration of the patient's viewpoint and reinforcing a shared focus on anticancer treatment options. When this focus became overwhelming, medical oncologists appeared to struggle to temper it, which in turn made balancing hope and realism even more challenging in subsequent interactions. This dynamic created a feedback loop, whereby each theme amplified the others, sustaining the cycle over time.

## Discussion

In this qualitative observational study, we developed the concept of a self-reinforcing cycle within decision-making consultations that may impede shared decision-making and, consequently, person-centered palliative care. This cycle emerged from four themes: the medical oncologist is balancing between hope and realism, there is little room for bad news, the medical oncologist's medical perspective is leading in medical decision-making, and the patient and medical oncologist have a shared focus on anticancer treatment, often with insufficient attention to the patient's personal context. While these themes are consistent with phenomena reported in previous literature [22,23,36], this study is the first to propose that these interactions collectively form a self-reinforcing cycle.

Several psychological and interactional processes likely underpin the four themes and the self-reinforcing cycle, including clinicians' psychology, patients' psychology, and their interactional dynamics—for example, collusions between patients and clinicians [37,38]. Understanding how these processes interact, as emphasized in, e.g., (medical) psychology

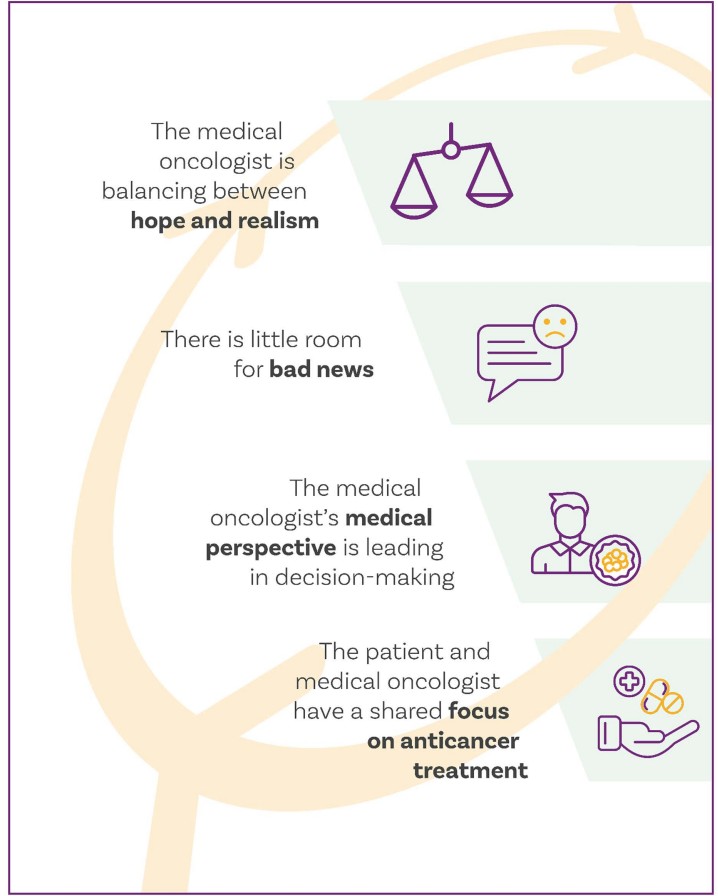

**Fig 1. Visual representation of the self-reinforcing cycle among themes.**

and behavioral science, is crucial and warrants further study, with our observations providing an important initial step toward this goal.

Several limitations should be considered. First, shared decision-making is dynamic and continuous, yet our observations were limited to consultations in which patients received scan results. Consequently, we could not account for earlier discussions between medical oncologists and patients that may have influenced these decisions. Nonetheless, consultations where patients receive scan results are pivotal in initiating the decision-making process regarding whether to continue, discontinue or start treatment. Second, our research team did not include expertise in (medical) psychology or behavioral science, disciplines that have strongly informed medical communication education and could have provided additional perspectives on the psychological and interactional mechanisms underlying the created themes and self-reinforcing cycle. Furthermore, medical education expertise, particularly in medical communication, could have enriched our analysis by illuminating how communication skills are taught, learned, and enacted in practice. We therefore recommend that future research integrate these disciplines alongside clinical and communication-focused expertise, enabling a more comprehensive understanding of the process shaping shared decision-making in oncology consultations.

Despite these limitations, our findings are particularly relevant considering the increasing complexity of shared decision-making due to the expansion of oncological treatment options over the past few decades [39]. Although the median survival gain has remained limited, substantial benefits can still be achieved for individual patients, placing

medical oncologists in an even more challenging position of balancing hope with realism. Furthermore, the importance of our findings is also underscored by our hypothesis that this cycle extends beyond oncology and may also affect persons with other life-limiting chronic illnesses, such as COPD, heart failure, renal failure, and dementia. This hypothesis is supported by an observational study [40] that showed a significant portion of patients with cancer and other life-limiting illnesses continued to receive hospital treatment in their final year. Approximately one-third were still hospitalized in the last month of life, despite most people in the Netherlands expressing a preference to die at home. This suggests a prevailing focus on life-extending treatment.

Follow-up to our study offers a valuable opportunity to integrate research with clinical practice and education. Audio- or video-recorded consultations can be reviewed to provide feedback to healthcare professionals, either collaboratively with patients, as demonstrated in studies on medically unexplained symptoms [41], or internally during team review sessions. Previous research shows that patient-collected recordings with structured feedback can enhance physicians' attention to contextual factors, improve outcomes, reduce hospitalizations, and lower healthcare costs [42]. Beyond feedback, supervision offers a promising avenue for addressing the psychological and interactional dynamics, including mechanisms such as collusion and the psychology of patients and clinicians [37,38]. Further research is needed to determine whether these approaches can effectively disrupt the cycle and strengthen shared decision-making within palliative care. Ultimately, such insights could inform clinical guidelines and medical education, better preparing healthcare professionals for person-centered consultations.

## Conclusion

In summary, our study suggests that in decision-making conversations, a self-reinforcing cycle of focus on treatment and hope may hinder shared decision-making and person-centered palliative care. This cycle is likely not confined to oncology alone. Further research is essential to explore the transferability of this cycle to other clinical contexts and determine whether our suggestions can effectively address it and improve care outcomes.

## Supporting information

**S1 Table. Study team and reflexivity.**
(DOCX)

**S2 Table. OPTION-12 items.**
(DOCX)

**S3 Table. Theme and category presence across consultations.**
(DOCX)

**S4 Table. Patient characteristics.**
(DOCX)

**S5 Table. Overview of consultation context.**
(DOCX)

**S6 Table. Codebook.**
(DOCX)

## Acknowledgments

We are grateful to all patients and medical oncologists/fellows who participated in our study. We would like to thank J. van Meurs for collecting the data of this study.



## Author contributions

**Conceptualization:** Daisy JM Ermers, Yvonne Engels, Henk J Schers, Kris CP Vissers, Evelien JM Kuip, Marieke Perry.

**Data curation:** Daisy JM Ermers.

**Formal analysis:** Daisy JM Ermers, Yvonne Engels, Marieke Perry.

**Investigation:** Daisy JM Ermers.

**Methodology:** Daisy JM Ermers, Yvonne Engels, Marieke Perry.

**Project administration:** Daisy JM Ermers.

**Supervision:** Yvonne Engels, Henk J Schers, Kris CP Vissers, Marieke Perry.

**Visualization:** Daisy JM Ermers.

**Writing – original draft:** Daisy JM Ermers.

**Writing – review & editing:** Yvonne Engels, Henk J Schers, Kris CP Vissers, Evelien JM Kuip, Marieke Perry.

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
