## [Decision Letter · Decision Letter 0]

31 Jul 2025

Dear Dr. Ermers,

We look forward to receiving your revised manuscript.

Kind regards,

Caroline Watts, PhD

Academic Editor

PLOS ONE

Journal Requirements:

“D.J.M. Ermers was supported by a personal grant from the Radboudumc. The funders had no role in study design, conduct, analysis, or reporting.”

3. In the online submission form, you indicated that [The sensitivity and confidentiality of the raw qualitative data makes sharing of the data without compromising confidentiality and privacy impossible, therefore consent for sharing of the raw data was not asked from the participants. Where possible, the raw qualitative data will be anonymized by data aggregation to enable sharing for reuse. The anonymous data used for analysis are available from the corresponding author and the research unit in CC (Onderzoek.anes@radboudumc.nl) upon reasonable request.].

5. We note that Figure 1 in your submission contain copyrighted images. All PLOS content is published under the Creative Commons Attribution License (CC BY 4.0), which means that the manuscript, images, and Supporting Information files will be freely available online, and any third party is permitted to access, download, copy, distribute, and use these materials in any way, even commercially, with proper attribution. For more information, see our copyright guidelines: http://journals.plos.org/plosone/s/licenses-and-copyright.

Reviewers' comments:

Reviewer's Responses to Questions

**Comments to the Author**

1. Is the manuscript technically sound, and do the data support the conclusions?

Reviewer #1: Partly

Reviewer #2: Yes

2. Has the statistical analysis been performed appropriately and rigorously?

Reviewer #1: N/A

Reviewer #2: N/A

3. Have the authors made all data underlying the findings in their manuscript fully available?

Reviewer #1: Yes

Reviewer #2: No

4. Is the manuscript presented in an intelligible fashion and written in standard English?

Reviewer #1: Yes

Reviewer #2: Yes

Reviewer #1: This is a relevant and timely study. However, I have a few major concerns with the manuscript: 1) The applied Reflexive Thematic Analysis (of what consists this method) and the way the researchers proceeded with this method is not or poorly described; 2) Backgound of the researchers: "meaningful communication"? This is not a profession like the professions of the other researchers (palliative care, medical oncology, primary care), what is the professional background?; 3) Reading the discussion it clearly lacks expertise in and input from liaison psychiatry/psycho-oncology: why did the researcher not associate a professional with this background?; 4) Why do the authors describe the questionnaires? This is a qualitative study, and there is no articulation between the quantitative and qualitative data; 5) The discussion seems to me rather superficial and this is a shame, since the study is very interesting; to give a few examples: a) SDM is not the only objective of a consultation, crtitique towards the oncologists have therefore be attenuated, and SDM should be placed within the context of all other objectives (e.g., support); b) some phenomena such as the shared focus on anticancer treatment can be explained by collusion (see the recent reviews and studies by Stiefel et al.), and if collusion is at work, simply feedback the observation to oncologists will not do it (see the beforementioned (for approaches, see these papers); c) other reactions of oncologists such as presenting bad news positively or responding to patient's negative emotions positively is understandable when one turns to the psychology of the clinician (e.g., feelings of guilt to harm the patient psychologically, fear of the patient's emotions - see for example the recent ESMO Guidelines on communication in chronic cancer care, in which an important part is dedicated to the "psychology of the clinician"; d) the interconnection of the four findings should also be discussed in light of the psychology of the patient, of the clinician and their interactional dynamics. a) - d) is important to discuss, since the suggested ways to improve the situation will depend on. Again, just to restore these finding to oncologists will not modify their communication behaviour, since it's not about technique but psychology; psycho-oncology has means to modify these beahviours (communication training, indicidual and group supervision, etc.)

Reviewer #2: The article is well written overall and outlines important applied research in shared decision-making. In order to help make the writing, analysis and findings clearer there are parts of the manuscript that would benefit from improvement as follows:

1. Introduction: Well-written overall.

"Vicious" cycle: the how underlying this is unclear to me. Discussion of this and elaboration in the introduction would be useful. Some discussion of the mechanisms by which this might happen might also be helpful for next steps in intervention.

2. Results & Discussion: The authors make points, such as "no room for bad news". It would be helpful to comment on literature and/or theoretical frameworks that suggest why this might be the case? Is there a wider societal context at play here? Or is it medical oncologists' own values? How might these intersect? Thinking about what these codes and findings might mean in terms of how we deliver care might deepen the insights provided by the manuscript.

3. The paper itself might benefit from consideration of theory or theoretical frameworks that elaborate on decision-making. What sort of shared decision-making models have already been suggested? How do the findings compare? Do they map onto similar domains or do they deviate significantly?

4. I would have liked to also understand a little more about the reflexive approach applied by the authors - what were the positions they approached the analysis with? What were their biases, strengths and limitations?

5. Lastly quantitative measures have been included in the paper write up but I'm not sure how these compare to the qualitative data. Making this clearer and how both sources of data have been analysed would help clarify the findings presented in the research.

.

Reviewer #1: No

Reviewer #2: No

---

## [Author Response · Author response to Decision Letter 1]

14 Jan 2026

Dear Editor,

As requested, please find below the direct links to the assets used from the Noun Project:

https://thenounproject.com/icon/chats-2055212/

https://thenounproject.com/icon/balance-1236636/

https://thenounproject.com/icon/cancer-6721448/

https://thenounproject.com/icon/treatment-6746479/

Please let me know if any additional information is required.

Kind regards,

Daisy Ermers

---

## [Editor Report · Decision Letter 1]

15 Mar 2026

Communication Patterns in Decision-Making Consultations between Patients with Advanced Cancer and Medical Oncologists: A Qualitative Observational Study

PONE-D-25-13553R1

Dear Dr. Ermers,

We’re pleased to inform you that your manuscript has been judged scientifically suitable for publication and will be formally accepted for publication once it meets all outstanding technical requirements.

Kind regards,

Caroline Watts, PhD

Academic Editor

PLOS One

Additional Editor Comments (optional):

Apologies for the delay with this paper. Thank you for your patience and consideration of reviewers comments.
---

## [Editor Report · Acceptance letter]

PONE-D-25-13553R1

PLOS One

Dear Dr. Ermers,

I'm pleased to inform you that your manuscript has been deemed suitable for publication in PLOS One. Congratulations! Your manuscript is now being handed over to our production team.

Kind regards,

on behalf of

Dr. Caroline Watts

Academic Editor

PLOS One